# The Potential of a Saliva Test for Screening of Alveolar Bone Resorption

**DOI:** 10.3390/healthcare11131822

**Published:** 2023-06-21

**Authors:** Yuichi Ikeda, Otofumi Chigasaki, Koji Mizutani, Yoshiyuki Sasaki, Norio Aoyama, Risako Mikami, Misa Gokyu, Makoto Umeda, Yuichi Izumi, Akira Aoki, Yasuo Takeuchi

**Affiliations:** 1Department of Periodontology, Graduate School of Medical and Dental Sciences, Tokyo Medical and Dental University (TMDU), Tokyo 113-8549, Japan; 2Tsukuba Health-Care Dental Clinic, Tsukuba 305-0834, Japan; 3Clinical Dental Research Promotion Unit, Faculty of Dentistry, Tokyo Medical and Dental University (TMDU), Tokyo 113-8549, Japan; 4Department of Periodontology, Kanagawa Dental University, Yokosuka 238-8580, Japan; 5Department of Periodontology, Osaka Dental University, Osaka 540-0008, Japan; 6Oral Care Periodontics Center, Southern Tohoku General Hospital, Southern Tohoku Research Institute for Neuroscience, Koriyama 963-8052, Japan; 7Department of Lifetime Oral Health Care Sciences, Graduate School of Medical and Dental Sciences, Tokyo Medical and Dental University (TMDU), Tokyo 113-8510, Japan

**Keywords:** alveolar bone loss, logistic models, mass screening

## Abstract

Oral health screening is important for maintaining and improving quality of life. The present study aimed to determine whether patients with a certain level of alveolar bone resorption could be screened by salivary bacterial test along with their background information. Saliva samples were collected from 977 Japanese patients, and the counts of each red-complex, that is, *Porphyromonas gingivalis*, *Treponema denticola*, and *Tannerella forsythia*, were measured using quantitative polymerase chain reaction analysis. Mean bone crest levels (BCLs) were measured using a full-mouth periapical radiograph. Multiple logistic regression analysis was used to determine associations between BCLs (1.5–4.0 mm in 0.5 mm increments) and explanatory variables, such as the number of each red-complex bacteria and the patients’ age, sex, number of teeth, stimulated saliva volume, and smoking habits. When the cutoff BCL value was set at 3.0 mm, the area under the curve, sensitivity, and specificity values were optimal at 0.86, 0.82, and 0.76, respectively. In addition, all tested explanatory variables, except sex and *T. denticola* count, were significantly associated with BCLs according to a likelihood ratio test (*p* < 0.05). Additionally, the odds ratio (OR) was substantially increased when a patient was >40 years old and the bacterial count of *P. gingivalis* was >10^7^ cells/µL (OR: >6). Thus, *P. gingivalis* count and patients’ background information were significantly associated with the presence of a certain amount of bone resorption, suggesting that it may be possible to screen bone resorption without the need for radiography or oral examination.

## 1. Introduction

Tooth loss not only deteriorates oral functions such as eating and speaking but possibly has a negative impact on systemic condition [1]. Various oral diseases can be caused by tooth loss; of these, periodontitis is the primary cause of tooth loss in adults and older people: it causes loss of attachment and subsequent alveolar bone resorption around tooth inflammation. According to the results of the Global Burden of Disease study in 2019, 1.1 billion people worldwide suffered from severe periodontitis, and their prevalence is still increasing [2]. The progression of periodontitis is generally not so noticeable, and the disease tends to be left untreated. At the time the patient notices its symptoms, periodontitis progresses to a severe stage, and extraction of the tooth is unavoidable in many cases. Furthermore, recent studies have revealed the association of periodontitis with systemic diseases, such as cardiovascular disease, diabetes, pneumonia, adverse pregnancy outcomes, etc. [3,4]. So, early detection of the disease’s signs, such as progressed bone resorption through dental checkups, is important to maintain good oral condition and systemic health.

Previous studies have shown that attachment loss values provide useful information on tooth survival [5] and that alveolar crest levels are associated with future tooth loss [6]. However, these examinations are time- and resource-consuming and are unsuitable to apply for large-scale oral health screening in epidemiological examinations [7]. For a definitive diagnosis of attachment loss, examining the entire circumference of the tooth with a periodontal probe is necessary. Alveolar bone resorption is also difficult to evaluate without the use of radiographic images by periapical/panoramic dental X-ray or computed tomography, and they should be taken by dental professionals. Moreover, there are concerns about frequent radiation exposure, even if it was performed for screening tests.

Screening methods must be applicable rapidly for the presumptive identification of unrecognized diseases or defects [8]. Many screenings have been developed for the provisional identification of periodontitis. At present, the community periodontal index (CPI) proposed by the WHO is commonly used worldwide as an oral health screening method [7]. This index was revised in 2013 and is often used to assess periodontal status, and probing pocket depth and bleeding on probing are measured and scored for all teeth [7,9]. This method is beneficial for easily estimating the periodontal condition as a score. However, the measurement of attachment loss is in sextant units, whereas periodontal pockets and gingival bleeding are in single-tooth units, and thus cannot accurately represent the attachment loss of an individual. Therefore, it is desirable to develop a simpler and more useful method for measuring attachment loss. Screening methods using oral bacteria or host-derived biomarkers in saliva have also been tried for identifying periodontitis. Although the analyses were performed based on a limited number of samples, the following items were proposed as biomarkers for periodontitis in recent studies; soluble CD40L [10], superoxide dismutase, SIRT-2 [11], IL-1ß, IL-6, MMP-8, MIP-1α [12], procalcitonin [13], Toll-like receptor-4, IL-18, uric acid [14] *Porphyromonas gingivalis*, and *Tannerella forsythia* [15]. However, few studies have targeted periodontal bone level as the outcome of the screening test.

In our previous studies, the prevalence and abundance of red-complex bacteria (i.e., *Porphyromonas gingivalis*, *Tannerella forsythia*, and *Treponema denticola*) in saliva, as well as age and smoking habits, were significantly associated with alveolar bone resorption [16,17]. Therefore, we hypothesized that combining a quantitative evaluation of red-complex bacteria in saliva and the background/clinical data of patients would enable a simple estimation of bone resorption levels. A simple test based on a questionnaire and saliva test could be applicable for large-scale screening for evaluating oral health risk.

The purpose of this study was to develop a method to screen for resorption above a certain amount of alveolar bone in a manner that does not involve radiation exposure and to determine whether screening at the alveolar bone level can be adapted for mass screening of periodontal disease.

## 2. Materials and Methods

### 2.1. Study Population

Precise information about the study population has been provided in previous studies [16,17]. Briefly, the present study included 977 patients who had an initial visit to a private dental clinic (Tsukuba Health-Care Dental Clinic) in Tsukuba, Japan from March 2003 to March 2006 and who requested and paid for a bacterial test. These individuals had little or no previous experience with periodontal treatment. The medical records of these individuals were provided for this study as retrospective data. The exclusion criteria for patients were as follows: <18 years old, use of antibiotics within the last 3 months, and a history of periodontal treatment within the last 6 months. The notification of the use of already existing medical records for research purposes was carried out by posting a document for comprehensive consent in the hospital, as stipulated by the laws and regulations of our country.

### 2.2. Clinical Evaluation

Full-mouth periapical radiographs were obtained from patients; the number of residual teeth was counted, and the vertical linear distances from the cementoenamel junction to the point of bone–root contact (bone crest level (BCL)) at the mesial and distal sites of each tooth were measured on the radiographic images by seven experienced periodontists (Y.I.(Yuichi Ikeda), O.C., K.M., Y.S., N.A., A.A. and Y.T.) using a measuring ruler previously described by Schei et al. [18]. The intraclass correlation coefficient was calculated by taking measurements BCL of 12 randomly selected periapical radiographs by each examiner, and the value was 0.756 (95% confidence interval (CI): 0.61–0.87). The mean BCL for all teeth except the third molar was calculated for each patient. Self-reports of current and previous smoking status were obtained from the patients. In addition, they were asked to chew paraffin gum for 5 min, after which stimulated whole saliva was collected and the salivary flow rate was recorded.

### 2.3. Bacterial DNA Extraction

Bacterial genomic DNA was extracted from the saliva samples using a High Pure PCR Template Preparation Kit (Roche, Basel, Switzerland) according to the manufacturer’s instructions. Briefly, 500 µL of saliva from each patient was washed with phosphate-buffered saline twice and centrifuged. Bacterial gDNA was extracted from the resultant pellet using the kit and stored at −30 °C until further analysis.

### 2.4. Real-Time Polymerase Chain Reaction

Real-time polymerase chain reaction (PCR) was conducted to quantify the amounts of three periodontopathic bacteria (*P. gingivalis*, *T. denticola*, and *T. forsythia*) using a LightCycler^®^ system (Roche Molecular Biochemicals, Mannheim, Germany) and Light-Cycler^®^ DNA Master SYBR Green I (Roche Molecular Biochemicals, Mannheim, Germany). Sequences of species-specific primers were used based on the 16s rRNA gene as previously described [19] (Appendix A). Amplification was performed in a 20 μL final volume containing 2 μL of template DNA, 2 μL of PCR Master Mix, 1 μM of each primer, and 4 mM of magnesium chloride. Detailed real-time PCR settings were previously provided by Chigasaki et al. [16]. Bacterial counts were collected from the stimulated saliva volume, and data were analyzed using LightCycler^®^ analysis software (Roche); the cutoff value for positivity was set at 1000 counts/mL per sample.

### 2.5. Statistical Analysis

Multiple logistic regression models were used to estimate BCLs using potential predictors, including patient background variables (age, sex, and smoking habits), clinical variables (number of teeth and stimulated saliva volume), and bacterial counts of the three red-complex bacteria in the saliva samples. BCL (1.5–4.0 mm measured in 0.5 mm increments) was used as the dependent variable. Explanatory variables were categorized into multiple groups as follows: age (<30, 30–39, 40–49, and >50), sex, number of teeth (<23 and ≥23), stimulated saliva volume (<5, 5–15, and ≥15 mL), and bacterial count (<10^5^, 10^5^–10^7^, and ≥10^7^). Sensitivities, specificities, likelihood ratios, and ROC curves were used to evaluate screening characteristics as described in the codification [20]. To visualize the ability to predict BCLs using the potential predictors, a receiver operating characteristic (ROC) curve table was created, and an ROC curve was plotted. The sensitivity, specificity, positive likelihood ratio (LR+), and negative likelihood ratio (LR−) were calculated, and the predictive performance of the model was evaluated according to the area under the ROC (AUC). JMP 9.0.3 statistical software (SAS Institute Inc., Cary, NC, USA) was used to conduct these analyses, with *p*-values of <0.05 considered significant.

## 3. Results

### 3.1. Clinical Data of the Study Population

In total, 324 male and 653 female patients were included in this study. As shown in Table 1 and Appendix A, the mean age of the patients was 38.0 ± 10.3 years, the mean number of residual teeth was 26.7 ± 2.3 (>90% of the patients had >24 residual teeth), and the mean BCL was 2.1 ± 1.2 mm.

### 3.2. Multiple Logistic Regression Analysis for Estimating BCL Using Clinical Parameters

The ROC curve was plotted at each cutoff value of the BCL from 1.5 to 4.0 mm in 0.5 mm increments (Figure 1). The AUC (0.73–0.89) and sensitivity (0.56–0.82) values increased as the cutoff value of the BCL increased (Table 2). The specificity value was >0.65 (0.66–0.81), whereas the LR+ and LR− values were 2.09–4.63 and 0.22–0.56, respectively, at each incremental stage. However, after using standard methods with at least 10 outcomes for each included explanatory variable, overfitting of the data was observed when BCL values were 3.5 or 4.0 mm. Thus, these data were considered unsuitable for further analysis. The LR test showed that all explanatory variables, except for sex, bacterial count of *T. denticola*, and bacterial count of *T. forsythia*, had a statistically significant goodness of fit when the BCL was 3.0 mm (Table 3). Excluding the overfitting data, the highest AUC and lowest LR− were observed when the BCL was 3.0 mm; thus, the results of multiple logistic regression analysis with a 3.0 mm BCL cutoff are shown in Table 4. A marked increase in the calculated odds ratio (OR) was observed as patient age increased (OR: 4.68–22.85). In addition, the OR significantly increased when the number of remaining teeth was <23 (OR: 8.38). Furthermore, the OR markedly increased as the stimulated saliva volume decreased (OR: 13.66–18.92). A high OR was also associated with smoking habits (OR: 2.99), and the OR significantly increased as the bacterial count of *P. gingivalis* increased (OR: 2.03–5.20). In contrast, no significant differences were observed with either *T. denticola* or *T. forsythia*.

When the number of residual teeth was excluded from the explanatory variables and the BCL was ≥3.0 mm, the AUC, LR+, and LR− values were 0.84, 3.88, and 0.28, respectively (Appendix A). Some explanatory variables, such as smoking habits, stimulated saliva volume, and the bacterial count of *P. gingivalis*, still had a statistically significant goodness of fit at 3.0 mm, and a substantial increase in the OR was observed in patients who smoked, had decreased salivary volume, and had high *P. gingivalis* counts (Appendix A).

## 4. Discussion

In the present cross-sectional study, we found that a certain amount of bone resorption could be determined using only salivary bacterial counting and simple patient background information, that is, radiography and/or oral examination by a specialist was not necessary. Specifically, alveolar bone resorption was significantly associated with *P. gingivalis* and *T. forsythia* counts, as well as other clinical factors, except for sex. In addition, BCL as the dependent variable appeared to be the most suitable for analysis at 3.0 mm. Given that the results associated with BCLs of 3.5 and 4.0 mm are overfitted, those associated with a BCL of 3.0 mm included the highest AUC and sensitivity values, as well as the lowest LR− value. Furthermore, a BCL of 3.0 mm was significantly associated with all the parameters tested in this study, except for sex and *T. denticola* counts.

According to Oda et al. [21], the average root length of all teeth, excluding the distal surface of the second molars, is approximately 14 mm. Considering the anatomically normal connective tissue attachment pattern, a BCL of 3.0 mm may result in approximately 15% bone resorption; more than 15% bone loss would not present incidentally. Periodontal disease is one of the main causes of alveolar bone resorption in adults [22,23]. Associations between periodontal disease and various systemic diseases, such as diabetes [24], cardiovascular disease [24], Alzheimer’s disease [25], rheumatism [26], cancer [27,28], and others [22], have been reported. The present simple screening test would not only contribute to increased awareness about oral health and inspire people to see the dentist but would also give them the opportunity to think about their own systemic health; it may contribute to improving quality of life. Since the ultimate goal would be to apply the present method to large-scale oral health screening, teeth with dental prosthesis or occlusal trauma were not excluded from the present analysis. As mentioned above, marginal alveolar bone resorption in adults is mainly caused by periodontal disease but can be due to endodontic lesions, occlusal trauma, root fracture, and orthodontic and dental prosthesis. Furthermore, it is not possible to determine whether bone resorption is progressive or obsolete using the present screening method. Thus, the present results should be interpreted with caution in relation to the risk of future progression of alveolar bone resorption.

Various methods have been proposed for rapid and efficient screening and assessment of periodontal conditions. These include a questionnaire-based system [29,30], examination of specific markers, for example, occult blood and hemoglobin in saliva [31,32] and lactate dehydrogenase [33] and hepatocyte growth factor in oral rinses [34], counting the number of bacteria associated with periodontal disease [15,35,36], and detection of toxins produced by such bacteria [37]. Recently, optical coherence tomography [38] and ultrasonography [39] have also been explored as noninvasive image analysis methods for the detection of periodontal disease.

However, only a limited number of studies reported the discriminability of screening tests for the periodontal condition in a large number of subjects. Shimazaki et al. [32] conducted a salivary occult blood test on 1998 adults aged 40 years with at least 20 teeth. Subjects with 15% of teeth bleeding on probing or 1 tooth with 4 mm or deeper probing depth were defined as having a poor periodontal status. As a result, the subjects suspected to have poor periodontal conditions could be identified by this test, although the specificity and sensitivity were not high (sensitivity 0.71 and specificity 0.52). Nomura et al. [33] also measured the salivary hemoglobin and lactate dehydrogenase levels in 92 adults aged over 20 years (mean 50.03 years) and tried to screen for periodontitis; in their study, periodontitis was diagnosed according to the criteria of the Center for Disease Control and Prevention in partnership with the American Academy of Periodontology [40]. For the periodontal screening, the hemoglobin or lactate dehydrogenase levels test itself did not provide substantially more accuracy than the community periodontal index. However, interestingly, combining these two tests, when samples tested positive for both hemoglobin and lactate dehydrogenase, the positive predictive value was 91.7%. A systematic review concluded that (i) no single marker or combination of markers can predict the conditions in the oral cavity and (ii) taking clinical measurements is currently the most reliable assessment method [41]. In the present approach, multiple elements (the presence of specific periodontal bacteria in saliva and the background/clinical data of patients) were used for the assessment of alveolar bone level. Although laboratory work is required for bacteriological evaluation, it is possible to evaluate the BCL without direct contact with the patient, which makes the method simple, patient-friendly, and useful. Since we also assumed that for some patients, it might be practically difficult to count the number of residual teeth in their oral cavity themselves, the number of residual teeth might need to be excluded as an explanatory variable. When we excluded this variable from our analyses, smoking habits, salivary volume, and *P. gingivalis* count still had high ORs (Appendix A).

This study has a few limitations. First, there might be participant bias, though the sample size was sufficient. Most patients were aged 20–59, and 55.6% had never experienced tooth loss. Furthermore, the patients in the present study were either from a university town or lived close to the town. Socioeconomic status is a well-known risk factor for both oral and diseases, and this group may have been relatively health conscious. For example, some systemic diseases, such as diabetes, are known to increase the risk for periodontitis, and reportedly, the current prevalence of diabetes in Japan is approximately 12% [42]. While the presence of diabetes was excluded from the explanatory variables in our analysis because of its low prevalence in the study group (10 patients: approximately 1% of all patients). The effectiveness of the present screening method may vary among subjects with different social and environmental backgrounds.

Furthermore, dental implants are currently often used to replace missing teeth, but they were excluded from the present analysis. Just like periodontitis, the inflammatory destruction of soft and hard tissues around osseointegrated implants can be induced by bacterial infection. Many studies have shown the high prevalence and/or increased level of periodontopathic bacteria, including *P. gingivalis*, in peri-implantitis [43,44,45]. Further, microleakage at the implant-abutment connection allows bacterial penetration into the inside of the implant body, and it acts as a bacterial reservoir leading to inflammation in peri-implant tissues [46,47]. While early marginal bone resorption can occur in relation with the establishment of supraclavicular-attached tissue around the implants, the bacteriological factor would not play a major role in the situation. Besides, there is a report that implant placement of full-arch implant-prosthetic rehabilitation could safely be performed, even in patients with diabetes, and their marginal bone loss was comparable to those of healthy patients at 10 years follow-up [48]. It seems it would be useful if the present screening method could be applied to implants, and we will clarify this in a future study.

## 5. Conclusions

Within the limitations of this study, the presence of a certain amount of bone resorption could be predicted using a prediction model based on a salivary bacterial test and simple patient background information. It might be helpful for preliminary assessment of periodontal status when a thorough periodontal examination is not possible.

## Figures and Tables

**Figure 1 healthcare-11-01822-f001:**
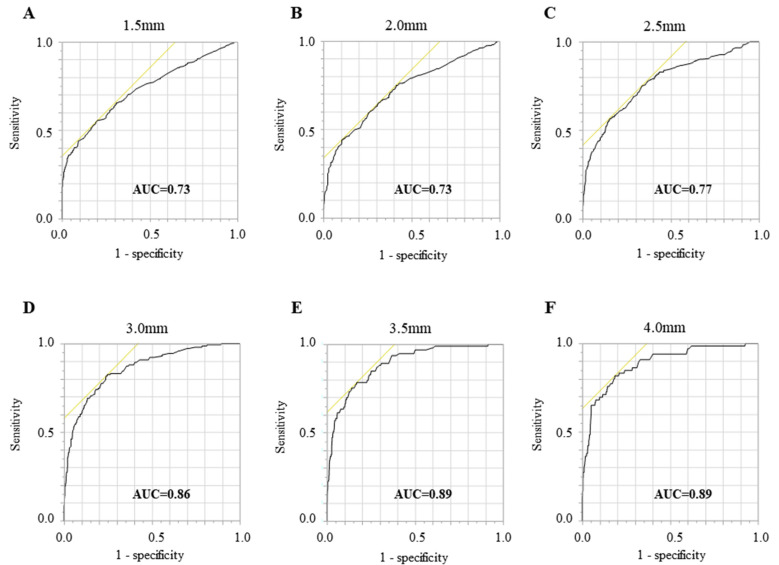
Receiver operating characteristic (ROC) curves for evaluating bone crest level by multiple clinical parameters. The ROC curves are plotted at the BCL (1.5–4.0 mm, in 0.5 mm increments) (**A**–**F**). The area under the ROC curve value is present at the bottom right of the figure.

**Table 1 healthcare-11-01822-t001:** Patient characteristics by age group (BOP, bleeding on probing; PD, pocket probing depth).

	Total	<30 Years	30–39 Years	40–49 Years	>50	*p*-Value
**Number of patients**	977	193	441	196	147	
**Sex**						0.356
Male	324	65	149	64	46	
Female	653	128	292	132	101	
**Smoking status**						0.179
Current smoker	162	41	73	29	19	
Non-smoker	815	152	368	167	128	
**Number of teeth**	26.7 ± 2.3	27.6 ± 1.0	27.2 ± 1.4	26.2 ± 2.3	24.5 ± 4.0	<0.001
**Bone crest level (mm)**	2.1 ± 1.2	1.6 ± 0.7	1.9 ± 0.9	2.4 ± 1.3	3.1 ± 1.4	<0.001
**BOP (%)**	24.3 ± 21.8	20.2 ± 21.0	22.9 ± 20.7	26.2 ± 23.2	31.3 ± 22.7	<0.001
**Site with PD ≥4 mm (%)**	15.2 ± 16.7	8.7 ± 12.1	13.4 ± 15.0	17.7 ± 18.4	25.9 ± 18.9	<0.001

**Table 2 healthcare-11-01822-t002:** Accuracy of screening at each cutoff point. BCL, bone crest level; Pos., positive; Neg., negative; Sn, sensitivity; Sp, specificity; LR+, a positive likelihood ratio; LR−, a negative likelihood ratio.

BCL	1.5 mm		2.0 mm		2.5 mm		3.0 mm		3.5 mm		4.0 mm	
	Pos.	Neg.	Pos.	Neg.	Pos.	Neg.	Pos.	Neg.	Pos.	Neg.	Pos.	Neg.
True	375	241	293	362	197	474	117	634	73	734	54	742
False	61	300	163	153	241	65	200	26	150	20	169	12
Sn	0.56		0.65		0.75		0.82		0.78		0.82	
Sp	0.80		0.69		0.66		0.76		0.83		0.81	
LR+	2.75		2.09		2.23		3.41		4.63		4.41	
LR−	0.56		0.51		0.37		0.24		0.26		0.22	
Notes	-		-		-		-		Overfitting		Overfitting	

**Table 3 healthcare-11-01822-t003:** Likelihood ratio test at each cutoff point. The explanatory variables are categorized into multiple groups. (*) indicates the statistically significant difference with a *p*-value of <0.05. LR, likelihood ratio; *P.g*, *Porphyromonas gingivalis*; *T.d*, *Treponema denticola*; *T.f*, *Tannerella forsythia*.

Explanatory Variables	1.5 mm		2.0 mm		2.5 mm		3.0 mm		3.5 mm		4.0 mm	
	χ^2^ for LR	*p*-Value	χ^2^ for LR	*p*-Value	χ^2^ for LR	*p*-Value	χ^2^ for LR	*p*-Value	χ^2^ for LR	*p*-Value	χ^2^ for LR	*p*-Value
Age	90.40	<0.001 *	74.37	<0.001 *	81.84	<0.001 *	59.69	<0.001 *	39.07	<0.001 *	25.67	<0.001 *
Sex	0.05	0.822	0.01	0.915	0.01	0.918	0.72	0.395	1.25	0.263	2.21	0.137
Number of teeth	10.44	0.001 *	15.20	<0.001 *	17.62	<0.001 *	33.27	<0.001 *	35.64	<0.001 *	30.96	<0.001 *
Stimulated saliva volume	1.80	0.407	1.16	0.561	1.51	0.470	16.27	<0.001 *	6.64	0.036 *	2.92	0.232
Smoking habit	7.16	0.008 *	9.31	0.003 *	6.76	0.009 *	14.64	<0.001 *	8.87	0.003 *	8.63	0.003 *
Log (bacterial count of *P.g* + 1)	28.96	<0.001 *	29.20	<0.001 *	30.42	<0.001 *	18.58	<0.001 *	26.80	<0.001 *	16.06	<0.001 *
Log (bacterial count of *T.d* + 1)	2.52	0.284	0.01	0.995	1.42	0.491	4.64	0.098	4.00	0.135	3.07	0.215
Log (bacterial count of *T.f* + 1)	1.14	0.567	0.13	0.935	0.09	0.955	1.84	0.398	0.04	0.983	0.51	0.774

**Table 4 healthcare-11-01822-t004:** Multiple logistic regression analysis at the cutoff point of the bone crest level at 3.0 mm. Adjusted odds ratios (ORs), 95% confidence interval (CI), and *p*-value were obtained from the multiple logistic regression analysis of the risk factor of periodontitis. Variables are categorized into multiple groups. (*) indicates the statistically significant difference with a *p*-value of <0.05. *P.g*, *Porphyromonas gingivalis*; *T.d*, *Treponema denticola*; *T.f*, *Tannerella forsythia*.

Variable	ORs	95% CI		*p*-Value
		Lower	Upper	
**Age**				
Less than 30	1.00 (reference)			
30 to 39	4.68	1.79	16.15	0.001 *
40 to 49	9.78	3.64	34.24	<0.001 *
Be equal to or more than 50	22.85	8.47	80.39	<0.001 *
**Sex**				
Male	1.00 (reference)			
Female	0.82	0.51	1.31	0.395
**Number of teeth**				
Equal to or more than 23	1.00 (reference)			
Less than 23	8.38	3.99	18.55	<0.001 *
**Stimulated saliva volume**				
Equal to or more than 15 mL	1.00 (reference)			
5 to 15 mL	13.66	2.75	249.88	<0.001 *
Less than 5 mL	18.92	3.53	355.31	<0.001 *
**Smoking habit**				
Nonsmoker	1.00 (reference)			
Current smoker	2.99	1.72	5.16	<0.001 *
**Log (bacterial count of *P.g* + 1)**				
Less than 5	1.00 (reference)			
5 to 7	2.03	1.28	3.23	0.003 *
Equal to or more than 7	5.20	2.22	12.17	<0.001 *
**Log (bacterial count of *T.d* + 1)**				
Less than 5	1.00 (reference)			
5 to 7	1.57	0.96	2.65	0.075
Equal to or more than 7	2.37	0.93	5.86	0.069
**Log (bacterial count of *T.f* + 1)**				
Less than 5	1.00 (reference)			
5 to 7	0.91	0.56	1.46	0.689
Equal to or more than 7	1.44	0.74	2.72	0.278

## Data Availability

Not applicable.

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
