# Peer review of "The Potential of a Saliva Test for Screening of Alveolar Bone Resorption"

_healthcare, 2023, doi:10.3390/healthcare11131822_

Round 1

Reviewer 1 Report

The authors, Ikeda et al, in their study, proposed a minimally invasive method to screen cases with alveolar bone loss using saliva. The authors used appropriate language with very limited language mistakes. However, there are some suggestions that the authors should take into consideration to improve the manuscript before proceeding to publication.

·         Line 29: we can not say AUC, sensitivity, and specificity, were optimal unless there is a scale to classify these measures into optimal, satisfactory, exceptional, and so on. And this scale should be mentioned clearly in the methods using appropriate citations.

·         Line 39 Introduction: The introduction needs to be restructured. For example, the authors should start talking about the problem by itself rather than defining "screening". Screening can be covered later in the introduction. My suggestion, the authors should talk about periodontitis/alveolar bone loss and its prevalence, complications, and problems associated with undiagnosed cases, if there is any study talked about the underestimation of bone loss and so on.

·         Then, screening of bone loss, and the drawbacks of any screening measures, if any (like cost, sensitivity, feasibility and so on). After that, saliva as a potential marker for oral diseases, papers that used saliva to detect oral conditions should be included such as DOI: 10.3390/jpm13020301 and others. More importantly, the significance of this study must be clarified. What kind of innovation or new knowledge can be provided and their clinical implications?

·         Line 83 study population: please add details about getting appropriate ethics approval/exemption to use patient samples especially because these patients attended private clinics not teaching hospitals.

·         Line 105 Bacterial DNA extraction: how was the saliva collected from patients, transported? did the authors use a special collection tube to prevent DNA degradation? A special section should be added for saliva collection. It should include: either saliva was stimulated or not stimulated, did the patients stop eating before collecting saliva? how many hours? tooth brushing and smoking before saliva collection, and so on

·         Line 129: why did the authors categorise the bacterial counts? why not continuous as continuous variables may provide more values to the results, especially because categorizing the bacterial counts like this is not based on scientific ground  The same applies to the patient's age and saliva volume

·         Line 141: because more than 90% of patients had more than 24 teeth, categorizing patients according to the number of teeth in >23 and <23 created huge differences in terms of the number of patients for this part in particular

·         I think TableS2 must be in the main text not supplementary, while Figure 1 can be supplementary

Author Response

Line 29: we can not say AUC, sensitivity, and specificity, were optimal unless there is a scale to classify these measures into optimal, satisfactory, exceptional, and so on. And this scale should be mentioned clearly in the methods using appropriate citations.

[Response]

We appreciate your kind advice. We have added citations to the codicil for the screening characterization.

Line 39 Introduction: The introduction needs to be restructured. For example, the authors should start talking about the problem by itself rather than defining "screening". Screening can be covered later in the introduction. My suggestion, the authors should talk about periodontitis/alveolar bone loss and its prevalence, complications, and problems associated with undiagnosed cases, if there is any study talked about the underestimation of bone loss and so on.

[Response]

We appreciate your kind advice. We have substantially rewritten the first half of the Introduction.

Then, screening of bone loss, and the drawbacks of any screening measures, if any (like cost, sensitivity, feasibility and so on). After that, saliva as a potential marker for oral diseases, papers that used saliva to detect oral conditions should be included such as DOI: 10.3390/jpm13020301 and others. More importantly, the significance of this study must be clarified. What kind of innovation or new knowledge can be provided and their clinical implications?

[Response]

We appreciate your kind advice. We have substantially rewritten the first half of the Introduction and cited more appropriate references.

Line 83 study population: please add details about getting appropriate ethics approval/exemption to use patient samples especially because these patients attended private clinics not teaching hospitals.

[Response]

We appreciate your kind advice. We added “The notification of the use of already existing medical records for research purposes was carried out by posting a document for comprehensive consent in the hospital, as stipulated by the laws and regulations of our country.” in 2.1. Study Population.

Line 105 Bacterial DNA extraction: how was the saliva collected from patients, transported? did the authors use a special collection tube to prevent DNA degradation? A special section should be added for saliva collection. It should include: either saliva was stimulated or not stimulated, did the patients stop eating before collecting saliva? how many hours? tooth brushing and smoking before saliva collection, and so on

[Response]

We appreciate your kind advice. Real-time PCR equipment is present in the clinic and saliva specimens are not transported. In addition, the collected saliva is frozen in a freezer for medical specimens at -30°C. The collection of stimulated saliva by chewing a paraffin block for 5 minutes is described in 2.2. Clinical Evaluation. The attending physician did not give any instructions to the patient regarding tooth brushing, eating, or smoking prior to saliva collection, and the times of tooth brushing and eating were not noted in the medical record.

Line 129: why did the authors categorise the bacterial counts? why not continuous as continuous variables may provide more values to the results, especially because categorizing the bacterial counts like this is not based on scientific ground  The same applies to the patient's age and saliva volume

[Response]

We appreciate your kind advice. Numerical measurement data can be used as quantitative data only when they are linear with the outcomes. Since bacterial count, age, and saliva volume are not linear with the outcome, BCL, it is a common procedure to transform them into categorical variables that are separated by inflection points.

Line 141: because more than 90% of patients had more than 24 teeth, categorizing patients according to the number of teeth in >23 and <23 created huge differences in terms of the number of patients for this part in particular

[Response]

We appreciate your kind advice. The number of remaining teeth was also not linear with the outcome, so it was separated by an inflection point of 23 teeth.

I think TableS2 must be in the main text not supplementary, while Figure 1 can be supplementary

[Response]

We appreciate your kind advice. Figure 1 and Table S2 were replaced according to your suggestion.

Reviewer 2 Report

This is a very large and impressive study. However, the paper has numerous shortcomings that should be fixed.

For starters, please apply the whole STROBE checklist to the fullest.

Also the introduction, the merit, and the conclusions should be improved. Many more rounds of review and revision may be necessary.

The sample size calculation with all its details should be mentioned.

Did you do qPCR for all those 977 patients?

Average

Author Response

This is a very large and impressive study. However, the paper has numerous shortcomings that should be fixed.

For starters, please apply the whole STROBE checklist to the fullest.

[Response]

We appreciate your kind advice. We checked the STROBE checklist. We uploaded it with a revised paper.

Also the introduction, the merit, and the conclusions should be improved. Many more rounds of review and revision may be necessary.

[Response]

W We appreciate your kind advice. e have made significant revisions to the introduction as a result of another reviewer's suggestion.

The sample size calculation with all its details should be mentioned.

[Response]

We appreciate your kind advice. Our study is a retrospective study, which means that already existing medical records were used for exploratory research purposes. Therefore, we believe that a sample size calculation is unnecessary. The sample size is determined only by the reference period of the medical records.

Did you do qPCR for all those 977 patients?

[Response]

We appreciate your kind advice. From March 2003 to March 2006, 3651 adult patients were examined for the first time. Among them, 977 patients had PCR tests performed and whole jaw radiographs taken. This was added to the method.

The possible existence of a selection bias in favor of PCR testing was noted in the discussion,

Reviewer 3 Report

In the Methods section, you mention in the inclusion of 977 patients. Please elaborate as to how the sample size was determined.

The term “gender” is used instead of “sex.” Sex generally refers to an organism's biological sex, while gender usually refers to either social roles based on the sex of a person. Please clarify.

In the Discussion section, you mention the possibility of “participant bias.” This typically refers to the tendency of participants (subjects) in an experiment to consciously or subconsciously act in a way that they think the experimenter or researcher wants them to act. Are you instead referring to “selection bias,” where participants are recruited in such a way that proper randomization is not achieved (thereby failing to ensure that the sample obtained is representative of the population intended to be analyzed)?

The manuscript would benefit from professional English language editing.

Numerous grammatical issues are present throughout the manuscript. For example, the use of “a” in “a full mouth periapical radiograph” reflects a single radiograph. However, it appears that you are referring to multiple periapical radiographs to cover all teeth within the mouth. Another example is the inappropriate use of capitalization: “Besides, Dental implants are currently…”

The manuscript would benefit from professional English language editing.

Author Response

In the Methods section, you mention in the inclusion of 977 patients. Please elaborate as to how the sample size was determined.

[Response]

We appreciate your kind advice. Our study is a retrospective study, which means that already existing medical records were used for exploratory research purposes. Therefore, we believe that a sample size calculation is unnecessary. The sample size is determined only by the reference period of the medical records.

The term “gender” is used instead of “sex.” Sex generally refers to an organism's biological sex, while gender usually refers to either social roles based on the sex of a person. Please clarify.

[Response]

We appreciate your kind advice. We replaced gender with sex because it means biological sex.

In the Discussion section, you mention the possibility of “participant bias.” This typically refers to the tendency of participants (subjects) in an experiment to consciously or subconsciously act in a way that they think the experimenter or researcher wants them to act. Are you instead referring to “selection bias,” where participants are recruited in such a way that proper randomization is not achieved (thereby failing to ensure that the sample obtained is representative of the population intended to be analyzed)?

[Response]

We appreciate your kind advice. From March 2003 to March 2006, 3651 adult patients were examined for the first time. Among them, 977 patients had PCR tests performed and whole jaw radiographs taken. This was added to the method.

The possible existence of a selection bias in favor of PCR testing was noted in the discussion,

The manuscript would benefit from professional English language editing.

Numerous grammatical issues are present throughout the manuscript. For example, the use of “a” in “a full mouth periapical radiograph” reflects a single radiograph. However, it appears that you are referring to multiple periapical radiographs to cover all teeth within the mouth. Another example is the inappropriate use of capitalization: “Besides, Dental implants are currently…”

[Response]

We appreciate your kind advice. Due to the short time frame for resubmission, the English editing by a professional vendor was not completed in time. We will definitely have the manuscript edited by a professional editing service in the future, so we would appreciate your forgiveness this time.

Round 2

Reviewer 1 Report

The authors responded to the comments. I would recommend accepting the paper in the current form

Moderate editing of English language required

Reviewer 3 Report

Questions from the previous round have been satisfactorily addressed.

Professional language editing is highly recommended